

# Measuring online social bubbles

Dimitar Nikolov, Diego F.M. Oliveira, Alessandro Flammini and Filippo Menczer

Center for Complex Networks and Systems Research, Indiana University, Bloomington, IN, United States

## ABSTRACT

Social media have become a prevalent channel to access information, spread ideas, and influence opinions. However, it has been suggested that social and algorithmic filtering may cause exposure to less diverse points of view. Here we quantitatively measure this kind of social bias at the collective level by mining a massive datasets of web clicks. Our analysis shows that collectively, people access information from a significantly narrower spectrum of sources through social media and email, compared to a search baseline. The significance of this finding for individual exposure is revealed by investigating the relationship between the diversity of information sources experienced by users at both the collective and individual levels in two datasets where individual users can be analyzed—Twitter posts and search logs. There is a strong correlation between collective and individual diversity, supporting the notion that when we use social media we find ourselves inside "social bubbles." Our results could lead to a deeper understanding of how technology biases our exposure to new information.

## INTRODUCTION

The rapid adoption of the Web as a source of knowledge and a social space has made it ever more difficult for people to manage the constant stream of news and information arriving on their screens. Content providers and users have responded to this problem by adopting a wide range of tools and behaviors that filter and/or rank items in the information stream. One important result of this process has been higher personalization (*Mobasher, Cooley & Srivastava, 2000*)—people see more content tailored specifically to them based on their past behaviors or social networks. Recommendation systems (*Ricci et al., 2011*), for example, suggest items in which one is more likely to be interested based on previous purchases, past actions of similar users, or other criteria based on one's past behavior and friends. Search engines provide personalized results as well, based on browsing histories and social connections (*Google, 2009b*; *Google, 2009a*).

It is common for users themselves to adopt filters in their online behavior, whether they do this consciously or not. For example, on social platforms such as Facebook, a large portion of users are exposed to news shared by their friends (*Bakshy et al., 2012*; *Matsa & Mitchell, 2014*). Because of the limited time and attention people possess and the large popularity of online social networks, the discovery of information is being transformed

Corresponding author
Dimitar Nikolov,
dnikolov@indiana.edu

from an individual to a social endeavor. While the tendency to selectively expose ourselves to the opinion of like-minded people was present in the pre-digital world (*Hart et al., 2009*; *Kastenmüller et al., 2010*), the ease with which we can find, follow, and focus on such people and exclude others in the online world may enhance this tendency. Regardless of whether biases in information exposure are stronger today versus in the pre-digital era, the traces of online behavior provide a valuable opportunity to quantify such biases.

While useful, personalization filters—whether they are algorithmic, social, a combination of both, and whether they are used with or without user awareness—have biases that affect our access to information in important ways. In one line of reasoning, *Sunstein (2002)*, *Sunstein (2009)* and *Pariser (2011)* have argued that the reliance on personalization and social media can lead people to being exposed to a narrow set of viewpoints. According to this hypothesis, one's existing beliefs are reinforced because they are locked inside so-called "filter bubbles" or "echo chambers," which prevent one from engaging with ideas different from their own. Such selective exposure could facilitate confirmation bias (*Baron, 2000*; *Nickerson, 1998*) and possibly create a fertile ground for polarization and misinformed opinions (*Nyhan & Reifler, 2010*; *McKenzie, 2004*; *Stanovich, West & Toplak, 2013*; *Silverman, 2011*).

These concerns are borne out to varying degrees in online user behavior data. For example, on Facebook, three filters—the social network, the feed population algorithm, and a user's own content selection—combine to decrease exposure to ideologically challenging news from a random baseline by more than 25% for conservative users, and close to 50% for liberal users (*Bakshy, Messing & Adamic, 2015*). The same study however highlights the difficulty in interpreting measurements of diverse information exposure. The decrease in exposure is significant, but the random baseline represents a completely bias-free exposure, which may not occur in reality. Our exposure is biased both in our explicit choices of information sources and implicitly through homophily—our tendency to associate with like-minded friends. Each social media filter may mitigate or amplify these biases. The combination of filters on Facebook still allows for exposure to some ideologically challenging news. But how does this compare to other ways of discovering information?

In a different Facebook study, users, especially partisan ones, were more likely to share articles with which they agree (*An et al., 2014*). Similar patterns can be seen on other platforms. On blogs, commenters are several times more likely to agree with each other than not (*Gilbert, Bergstrom & Karahalios, 2009*), and liberals and conservatives primarily link within their own communities (*Adamic & Glance, 2005*). On Twitter, political polarization is even more evident (*Conover et al., 2011*; *Conover et al., 2012*). When browsing news, people are more likely to be exposed to like-minded opinion pieces (*Flaxman, Goel & Rao, 2013*), and to stay connected and share articles with others having similar interests and values (*Grevet, Terveen & Gilbert, 2014*). In the context of controversial events that are highly polarizing, web sources tend to be partial and unbalanced, and only a small fraction of online readers visit more than two different sources (*Koutra, Bennett & Horvitz, 2014*). To respond to such narrowing of online horizons, researchers have started to concentrate

on more engaging presentation of disagreeable content (*Doris-Down, Versee & Gilbert, 2013*; *Munson & Resnick, 2010*; *Graells-Garrido, Lalmas & Quercia, 2013*).

In domains outside of political discourse there is less evidence that personalization and social networks lead to filter bubbles. Recommendation systems have a diversifying effect on purchases (*Hosanagar et al., 2013*), and search engines have had a democratizing effect on the discovery of information, despite the popularity-based signals used in their ranking algorithms (*Fortunato et al., 2006*).

Aspects of the filter bubble hypothesis have so far been quantified for specific platforms like blogs (*Adamic & Glance, 2005*), Facebook (*Bakshy, Messing & Adamic, 2015*), and Twitter (*Conover et al., 2011*), but not across different classes of information sources. Indeed, social media and blogs could be very different from other types of sites, because of the strong social influence in them. What these differences may be and how they affect information consumption is an open question. For example, on the one hand, one would imagine homophily to contribute to the formation of echo chambers in social networks. On the other hand, the abundance of weak ties between individuals in different communities (*Bakshy et al., 2012*) could lead to highly diverse exposure. In this study we look at the diversity of information exposure more broadly. Our goal is to examine biases inherent in different types of online activity: information search, one-to-one communication from email exchanges, and many-to-many communication captured from social media streams. *How large is the diversity of information sources to which we are exposed through interpersonal communication channels, such as social media and email, compared to a baseline of information seeking?* We answer this question at the *collective* level by analyzing a massive dataset of Web clicks. In addition, we investigate how this analysis relates to the diversity of information accessed by *individual* users through an analysis of two additional datasets—Twitter posts and search logs. Figure 1 illustrates our empirical analysis: we measure how the visits by people that are engaged in different types of online activities are distributed across a broad set of websites (Figs. 1A and 1C) or concentrated within a few (Figs. 1B and 1D).

We carry out our analyses on all web targets as well as on targets restricted to news sites. The latter are of particular relevance when examining bias in public discourse. We do not make any additional distinctions regarding the type of content people visit, such as opinion pieces versus reporting, or differing ideological slant. We do not consider beliefs, past behaviors, or specific interests of information consumers. These deliberate choices are designed to yield quantitative measures of bias that do not depend on subjective assessments. Our results are therefore general and applicable to different topics, geographical regions, interests, and media sources.

## METHODS

To study the diversity of information exposure we use a massive collection of Web clicks as our primary dataset, and two supplementary datasets of link shares on Twitter and AOL search clicks. Code is available to reproduce our entire data processing procedure, described next (https://github.com/dimitargnikolov/web-traffic-iu).

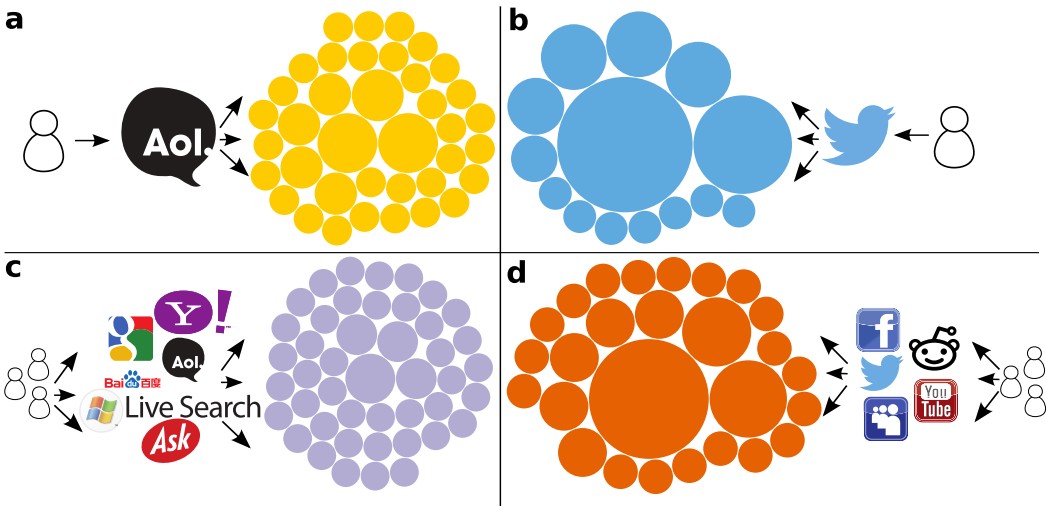

**Figure 1 Diversity of information sources accessed through different online channels.** Each circle represents a unique website, and its area is proportional to the number of pages accessed on that website. (A) Links clicked by a single search engine user. (B) Links shared by a single Twitter user. (C) Search traffic generated by a collection of users. (D) Social media traffic generated by a collection of users. In each case, a random sample of 50 links was taken for a period of one week. These examples illustrate typical behaviors gleaned from our data. On the left we see more heterogeneous patterns with search traffic distributed more evenly among several sources (higher Shannon entropy $H_a = 5.1$ and $H_c = 5.4$). The patterns on the right are more homogeneous, with fewer sources dominating most social traffic (lower entropy $H_b = 3.1$ and $H_d = 4.2$).

## Click dataset

The click data we use comes from a publicly available dataset collected at the edge of the Indiana University network (*Meiss et al., 2008*), which allows us to obtain a trace of web requests (http://cnets.indiana.edu/groups/nan/webtraffic/click-dataset/). Each request record has a target page, a referrer (source) page, and a timestamp. Privacy concerns prevent any identifying information about individual users or clients from being collected, making it impossible to associate any request with any particular computer or person. We only use the traffic coming from self-identified browsers to filter out search engine crawlers and other bots. The data only includes traffic originating inside the Indiana University network and requesting external pages.

This collection draws from a diverse population of over 100 thousand people, and spans a period of 41 months between October 2006 and May 2010.

Since in the click data it is not possible to distinguish with full certainty requests resulting from human clicks and requests auto-generated by the pages, we filter out any requests for files other than web pages, such as JavaScript, images, video, and so on based on the file extension. This results in the shrinking of the dataset by a factor of 5. Since the file extension is not always present in the URL, this method is not guaranteed to remove all non-human click data. However, it provides a good first approximation of human clicks, and we further address this issue with additional data filtering described later in this section.

Once non-human traffic is removed from the dataset based on file extensions, the path in the URL is discarded and the resulting clicks are only identified by the referrer and

**Table 1** Most frequent sources for each category of traffic and their corresponding numbers of clicks.

| Search | Social media | Email |
|---|---|---|
| google.com (9,792,271) | facebook (241,286) | mail.yahoo.com (573,248) |
| search.yahoo.com (1,753,478) | reddit.com (75,706) | mail.live.com (255,226) |
| search.msn.com (552,294) | twitter.com (59,471) | mail.google.com (153,436) |
| bing.com (372,819) | myspace.com (49,710) | webmail.aol.com (106,278) |
| ask.com (247,314) | youtube.com (45,146) | hotmail.msn.com (52,325) |
| search.naver.com (110,748) | linkedin.com (8,177) | webmail.iu.edu (7,217) |

target domains. We take referrer and target domains as proxies for websites. This level of granularity allows us to address the research question while avoiding the problem of the sparseness of the traffic at the page level—users typically visit most pages once.

Even if we identify a domain with a website, not all sites are equal—*wikipedia.org* has more diverse content than *kinseyinstitute.org*. Furthermore, one needs to decide whether to represent domains at the second or higher level. In many instances, higher-level domains reflect infrastructural or organizational differences that are not important to measure diversity (e.g., *m.facebook.com* vs. *l.facebook.com*). In others cases, using second-level domains may miss important content differences (e.g., *sports.yahoo.com* vs. *finance.yahoo.com*). To address this issue, we performed our analysis using both second- and third-level domains. As discussed below, these analyses yield very similar results. In the remainder of the paper we consider second-level domains, but account for special country cases; for example, domains such as *theguardian.co.uk* are considered as separate websites. Once we have a definition of a website, we use the number of clicks in the data to compute a diversity measure as discussed below.

After extracting the domain at the end points of each click, we examined the most popular referrers in the dataset and manually assigned them to the *search*, *social media,* and *email* categories. We then filtered the click data to only include referrers from these categories. In addition, we excluded targets from these same categories, because we are specifically interested in the acquisition of new information. For example, activities such as refining searches on Google and socializing on Facebook are unlikely to represent such discovery.

Subsequent data filtering was performed to exclude other likely non-human traffic, such as traffic to ad and image servers, traffic resulting from game playing or using browser applications such as RSS readers, and traffic to URL shortening services. Since it is impossible to exclude all non-human traffic, we focused on filtering out those target domains that constitute a significant portion of overall traffic. We used an iterative procedure in which we examined the top 100 targets for each category and manually identified traffic that is non-human. This procedure was repeated until the list of top 100 domains in each category was composed of legitimate targets. Table 1 lists the top six referrers in each category.

The filtered dataset includes over **106 million records**, roughly representing someone clicking on a link from a search engine, email client, or social media site, and going to one of almost **7.18 million targets** outside these three categories.

**Table 2  Seed DMOZ categories for the crawler used to extract a list of close to 3,500 news sites.**

| |
|---|
| www.dmoz.org/News/Internet_Broadcasts/ |
| www.dmoz.org/News/Magazines_and_E-zines/ |
| www.dmoz.org/News/Newspapers/ |
| www.dmoz.org/News/Internet_Broadcasts/Audio/ |
| www.dmoz.org/Arts/Television/News/ |
| www.dmoz.org/News/Analysis_and_Opinion/ |
| www.dmoz.org/News/Alternative/ |
| www.dmoz.org/News/Breaking_News/ |
| www.dmoz.org/News/Current_Events/ |
| www.dmoz.org/News/Extended_Coverage/ |

## News targets in the click dataset

To measure diversity of information exposure in the context of news, we created a separate dataset consisting only of clicks to news targets. Due to the specific research question we are investigating, we believe it is important to build this dataset in an open and comprehensive way, including less popular news outlets. To this end, we extracted the list of news websites by traversing the DMOZ open directory (http://www.dmoz.org/) starting with the seed categories shown in Table 2 and crawling their subcategories recursively. Following the crawl, the list of news targets was filtered as follows.

1. Each URL was transformed to a canonical form and only the domain name was kept.
2. Domains falling in one of the predefined categories—search, social media and email—were removed. URLs from popular blogging platforms, Wiki platforms, and news aggregators were also removed (see Table 3).
3. An iterative filtering procedure was applied to remove targets of non-human traffic, such as from RSS clients, advertising, and content servers.

The above procedure resulted in nearly **3,500 news sites**. We used this list to filter the targets in the click collection, yielding the news dataset used in our analysis.

## Diversity measure and relationship to traffic

To quantify the diversity of an information source *s* we look at all targets reached from websites in category *s* and compute the Shannon entropy

$$H_s = -\sum_{t \in T(s)} p_t \log p_t,$$

where $T(s)$ is the set of target websites reached from referrer sites in *s*, and $p_t$ is the fraction of clicks requesting pages in website *t*.

Entropy (*Shannon, 1948*) is a measure of uncertainty over a set of possible outcomes. It is maximized when all outcomes are equally likely, and minimized when only a single outcome is likely (indicating full certainty). Used over the set of domain probabilities as we have done above, the entropy gives the uncertainty in the websites that will be

**Table 3** Blogging platforms, Wiki platforms and news aggregators filtered out of the list of news sites.

| Blogging platforms | Wiki platforms | News aggregators |
|---|---|---|
| blogger.com | wikipedia.org | news.aol.com |
| blogspot.com | wictionary.org | news.google.com |
| hubpages.com | wikibooks.org | news.yahoo.com |
| livejournal.com | wikidata.org | |
| tumblr.com | wikimediafoundation.org | |
| typepad.com | wikinews.org | |
| wordpress.com | wikiquote.org | |
| wordpress.org | wikisource.org | |
| xanga.com | wikiversity.org | |
| | wikivoyage.org | |

accessed given a category of referrers. By measuring diversity over a set of domains, our approach captures the intuition that visiting 10 pages (for example, news articles) from 10 different sites implies a more diverse exposure than visiting 10 pages from the same site. The implications of this assumption are further debated in the 'Discussion' section. We considered an alternative method of measuring diversity based on the Gini coefficient (*Sen, 1973*), and found the results discussed below to be robust with respect to the choice of diversity measure.

The traffic volume in our click dataset varies significantly over time and across the three categories, as shown in Fig. 2A. A similar pattern emerges for the dataset of news targets (see inset). These vast volume differences make it necessary to understand the relationship between traffic volume and the diversity of an information source. To do so, we measure the diversity over samples of increasing numbers of clicks. From Fig. 2B we see that the diversity measurements indeed depend on volume, especially for small numbers of clicks; as the volume increases, the diversity tends to plateau. However, the dependence of diversity on number of clicks is different for each category of traffic. Therefore, instead of normalizing each category of traffic by a separate normalization curve, we account for the dependence by using the same number of clicks. This makes our approach easier to generalize to more categories and datasets, since it does not require the fitting of a separate curve to each case. We compute the diversity over traffic samples of the same size (50,000 clicks per month for all targets, and 1,000 clicks per month for news targets) for each category in our analysis.

## Auxiliary datasets

In the second part of our analysis we make use of two auxiliary datasets to disentangle the relationship between collective diversity—as seen in the targets accessed by a community of users—and individual diversity—as seen in the targets accessed by a single user. From both datasets, we are able to recover a referring website, a target website, and an associated user identifier.

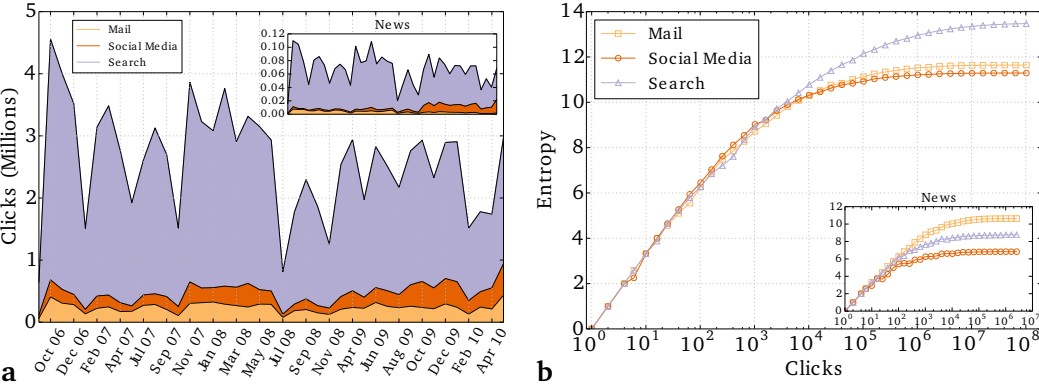

**Figure 2 Dependence of entropy on traffic volume.** (A) Traffic volume as a function of time for three different sources. (B) Entropy as a function of traffic volume. Error bars become negligibly small at 400 clicks, and are omitted for clarity. With fewer than 400 clicks, the entropy for the different categories is not significantly different. The insets show click volume and entropy for news traffic (same scale if not shown).

### *AOL search logs*

In the search dataset we have information about search engine sessions from a period of three months in 2006, containing over **18 million clicks** by over **half a million users** of the AOL search engine.

### *Twitter posts*

In the social media dataset we have a sample of almost **1.3 billion public posts** containing links shared by over **89 million people** on Twitter during a period of 13 months between April 2013 and April 2014. This data was obtained from the Twitter streaming API (https://dev.twitter.com/streaming/overview). We treat these records as proxies for clicks, assuming that users have visited the shared pages.

## RESULTS

Figure 3A presents our main finding: the diversity of targets reached from social media is significantly lower than those reached from search engine traffic, for all traffic as well as news targets (inset). This result holds for both second- and third-level domains, and is consistent with results obtained using an alternative measures of diversity. The observed differences in diversity did not change significantly over a period of three and a half years (see Fig. 3B). This empirical evidence suggests that social media expose the community to a narrower range of information sources, compared to a baseline of information seeking activities. Figure 4 illustrates the top targets of traffic from search and social media on a typical week. The diversity of targets reached via email also seems to be higher than that of social media, however the difference is smaller and its statistical significance is weaker due to the larger noise in the data. The difference in entropy is larger and more significant for traffic from email sources to news targets.

While we wish to ultimately understand the biases experienced by individuals, the diversity measurements based on anonymous traffic data do not distinguish between users, and therefore they reveal a *collective social bubble*, as illustrated in Figs. 1C and 1D. It is

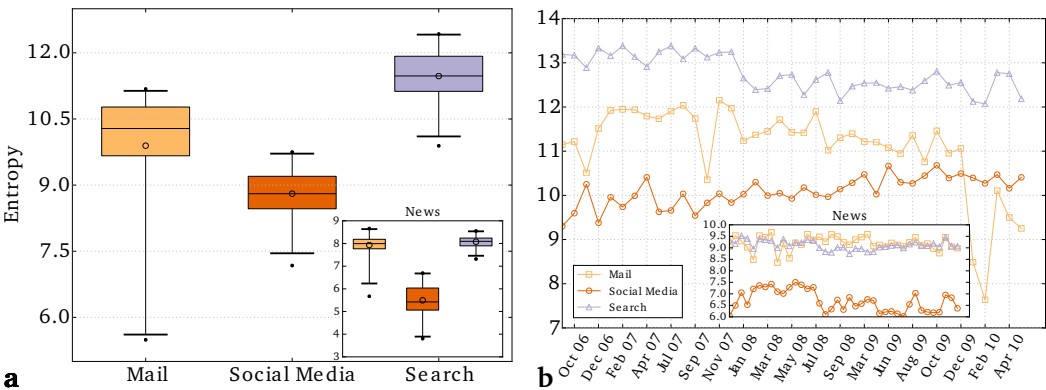

**Figure 3** **Diversity of sources accessed by different online activities.** (A) Overall entropy for different traffic categories over the full range of data (Oct 2006–May 2010). Each box represents the range of data between the 25th and 75th percentiles. The top and bottom whiskers show the 99th and 1st percentiles, respectively. The horizontal line and the hollow point inside each box mark the median and mean entropy, respectively. The filled points are outliers. The uncertainty was computed over data points representing the clicks that occurred over one calendar month. (B) Entropy as a function of time. We smooth the data by applying a running average over a three-month sliding window that moves in increments of one month. Error bars are negligibly small and thus omitted. The insets plot the entropy for news traffic (same scale if not shown).

at first sight unclear whether the collective bubble implies individual bubbles, or tells us anything at all about individual exposure. The number of clicks per user, or even the number of users could vary to produce different individual diversity patterns resulting in the same collective diversity. In theory, high collective diversity could be consistent with low individual diversity, and vice versa. Therefore we must investigate the relationship between collective and individual diversity measurements. To this end, we analyze the two auxiliary datasets where user information is preserved (see 'Methods'). For both datasets, we measure the diversity for individual users, and collectively disregarding user labels. The strong correlation between collective diversity and average user diversity (Fig. 5) suggests that our results relate not only to a collective bubble, but also to *individual social bubbles*, as illustrated in Figs. 1A and 1B.

## DISCUSSION

We have presented evidence that the diversity of information reached through social media is significantly lower than through a search baseline. As the social media role in supporting information diffusion increases, there is also an increased danger of reinforcing our collective filter bubble. A similar picture emerges when we specifically look at news traffic—the diversity of social media communication is significantly lower than that of search and inter-personal communication. Given the importance of news consumption to civic discourse, this finding is especially relevant to the filter bubble hypothesis.

Our results suggest that social bubbles exist at the individual level as well, although our evidence is based on the relationship between collective and individual diversity and therefore indirect. Analysis of traffic data with (anonymized) user identifiers will be necessary to confirm this conclusively. In addition, these results apply to the population of

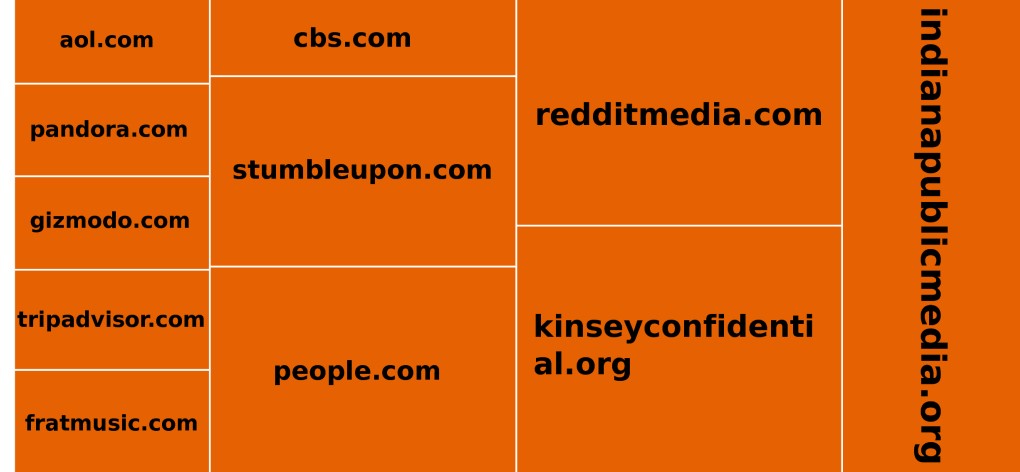

**Figure 4 Top websites that are targets of 40% of clicks for search (A) and social media (B).** This illustration refers to a typical week, with entropy close to (within one standard deviation from) average. The area of each rectangle is proportional to the number of clicks to that target. While these websites reflect the sample of users from Indiana University as well as the time when the data was collected, these contexts apply to both categories of traffic. Therefore the higher concentration of social media traffic on a small number of targets is meaningful.

users from Indiana University during the time period when the data was collected—from late 2006 to mid 2010. Repeating these experiments on other populations would be beneficial to establish the generality of our findings. Indeed, the social media and search landscapes have changed since 2010 and how that affects the diversity of information exposure for people is an interesting question for ongoing research.

Further research is also needed to tease out the influence of social versus algorithmic effects. Both are present in systems like Facebook—the algorithmic effect has to do with how a platform populates the feed for each user, which can be determined by a variety of individual and collective signals such as past social interactions and popularity. It seems

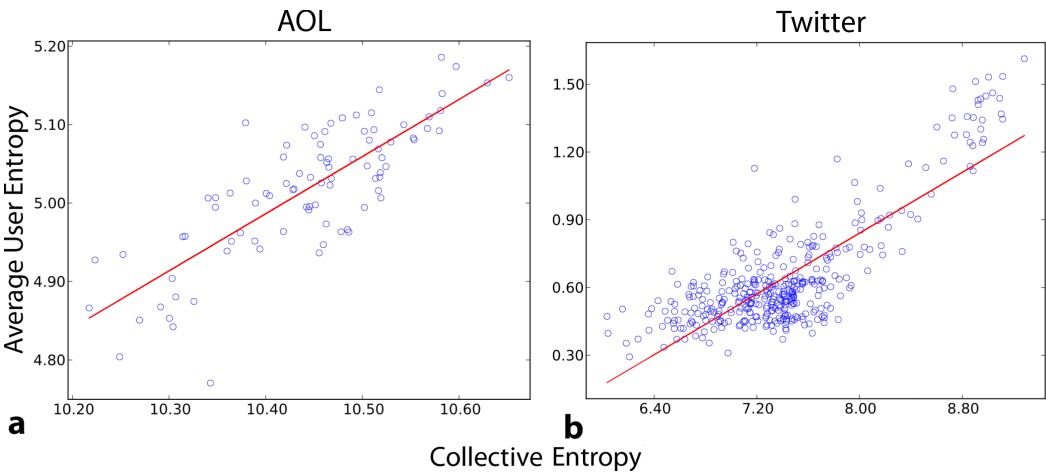

**Figure 5  Correlation between collective and average individual entropy.** Each point corresponds to an equal-size sample of links for each of a set of users sampled during a period of one day, to avoid volume bias in the entropy measurements. (A) Users sampled from search engine logs and their clicks (Pearson's $r = 0.8$). We sampled 60 clicks from each of 50 users per day. (B) Users sampled from Twitter and their shared links ($r = 0.8$). We sampled 10 links from each of 10,000 users per day.

unlikely that the relationship between algorithmic and social effects can be extracted from traces of online behavior as done here, without conducting controlled user studies.

These results also come with the caveat that in our analysis we do not try to quantify the diversity inside each domain. We are assuming that the diversity of content is higher across different domains than across the pages within a single domain. The problem of quantifying the diversity of the content inside a single domain is a significant research problem in its own right, and one that would greatly benefit this and similar lines of research. Quantifying domain diversity will likely need to be tackled by looking at the content of individual pages as other measures, such as the number of sub-domains or the number of pages inside the domain, are more indicative of size and popularity, but not necessarily of diversity.

Finally, in our study all social media traffic and all search traffic is merged. Further work is needed to tease out possible differences in diversity of information accessed through distinct search and social platforms. The social media platforms that exist today have important differences in functionality, and it will be worthwhile to investigate whether all these services under the umbrella of social media have similar properties when it comes to diverse information exposure.

## CONCLUSION

Our findings provide the first large-scale empirical comparison between the diversity of information sources reached through different types of online activity. The traffic dataset gives us a unique opportunity to carry out this analysis. We are not aware of any other methods, based on publicly available data, for contrasting different information access patterns produced by the same set of users, in the same time period.

While we have found quantitative support of online social bubbles, the question of whether our reliance on technology for information access is fostering polarization and

misinformation remains open. Even with ample anecdotal evidence (*Mervis, 2014*), we have yet to fully comprehend how today's technology biases exposure to information.

## ACKNOWLEDGEMENT

We are grateful to Mark Meiss for collecting the web traffic dataset used in this paper.

### Funding

This manuscript is based upon work supported in part by the James S. McDonnell Foundation and the National Science Foundation (award CCF-1101743). The funders had no role in study design, data collection and analysis, decision to publish, or preparation of the manuscript.

### Grant Disclosures

The following grant information was disclosed by the authors:
James S. McDonnell Foundation.
National Science Foundation: CCF-1101743.

### Competing Interests

Filippo Menczer is an Academic Editor for PeerJ Computer Science. The other authors declare there are no competing interests.

### Author Contributions

- Dimitar Nikolov and Diego F.M. Oliveira conceived and designed the experiments, performed the experiments, analyzed the data, wrote the paper, prepared figures and/or tables, performed the computation work, reviewed drafts of the paper.
- Alessandro Flammini and Filippo Menczer conceived and designed the experiments, analyzed the data, wrote the paper, reviewed drafts of the paper.

### Data Availability

The Click Collection System Dataset from the IU School of Informatics and Computing restricts the access and use of this dataset for research purposes only, and requires all interested parties to submit a formal request at http://cnets.indiana.edu/groups/nan/webtraffic/click-dataset/.

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
