# Peer review of "Measuring online social bubbles"

_PeerJ Computer Science, doi:10.7717/peerj-cs.38_

## Round 0.1 · original submission · Major Revisions

This is an interesting paper that ought to be published.

There are enough issues raised by reviewers that a resubmission is required. That said, I think this is on the "minor" end of the "major revisions" category.

The main points I want to see addressed are:
* more justification of several assumptions and choices made - for example, how does measuring bubbles in this way help the research community? why is entropy used? can you give a bit more explanation of what happened in the pre-Web days?
* thoroughly clean up typos (I found "born" -> borne" on p1, for example)
* reproducibility - PeerJ requires data to be publicly available; filtering process should be even better described
* Reviewer 2 is the most critical and makes several points all of which I would like you to address specifically.

Please do not send a DOC with tracked changes - a rebuttal letter is OK.

Reviewer 1 ·

Basic reporting

No Comments

Experimental design

Generally looks fine. The question is defined and the calculation process is explained. A single issue appears where the "density measure" is identified. However this measure and its calculation is presented clearly, but the paper fails to talk about the literature of this measure and the reason that the authors have chosen this method.

Validity of the findings

No comments

Additional comments

It is recommended that authors include the literature for the diversity measure. Then they can explain why the presented method is chosen.
Also more explanation regarding the importance of the study is required. Why quantifying the filter bubble hypothesis across different information sources is important and required?

Cite this review as

Reviewer 2 ·

Basic reporting

The key idea proposed in this article is that the diversity of information sources we are exposed to is significantly larger through search than through social media or e-mail communications. The explanation of the intuition behind this idea as well as the development of the argument is done well in the introduction.The article presents a good introduction and suitable background to demonstrate how this work fits in with related work.

Experimental design

Some of what the authors call intuitions and assumptions are not clearly explained - why does visiting 10 articles from 10 different news domains constitute greater diversity? Does this not depend on the type of domain? There is a mention of differences in domains such as wikipedia but the notion of greater diversity as it applies here is not clear. Similarly a more careful if only brief discussion on how excluding image and video search does or does not impact search results will be useful.

The discussion on sample volume and entropy seems a little incomplete. Why restrict sample sizes and not use some normalization measure instead? Search generates more data than emails for example and that is an important factor to consider. If the sample sizes were restricted, were multiple independent experiments conducted to validate findings?

Validity of the findings

There is considerable manual filtering involved in the data used for experiments in this article. Reproducing of results is of some concern in such cases. Can the authors make the subset of the data (used as input for this article after all the processing done on the original Click Dataset) available? Figure 3(a) shows results for a month. How representative is this of the entire data?

Cite this review as

Reviewer 3 ·

Basic reporting

Axis names for diagrams should be more informative. For instance, comparing Figure 5 data is not possible without reading the whole caption.

Experimental design

The article debates an important issue yet lacks proper debate on the significance and impacts of the matter that is being discussed. As I have commented on the paper in the attached file, the personalized source of information may not be the significant matter, as it has been always like this for people who had the preferences over them, for instance, left or right wing newspapers. What matters is the choice, whether or not an individual seeks for personalized source of news and entertainment or just is thrown at the main stream without knowing other choices exist. This should be considered as well in the introduction.

Validity of the findings

As I have commented in the PDF attached, some assumptions are questionable, such as increase in trusting the validity of news when it is reported from different sources.

Additional comments

No Comments.

Annotated reviews are not available for download in order to protect the identity of reviewers who chose to remain anonymous.
Cite this review as

---

## Round 0.2 · accepted · Accept

It seems ready to go - I don't see the typo mentioned by the reviewer.

Reviewer 2 ·

Basic reporting

The authors have made modifications that improve the clarity of the article.

Experimental design

No Comments

Validity of the findings

No comments

Additional comments

Typo: The caption on Figure 3 (a) reads (b)

Cite this review as